# Development of Anti-HIV Therapeutics: From Conventional Drug Discovery to Cutting-Edge Technology

**DOI:** 10.3390/ph17070887

**Published:** 2024-07-04

**Authors:** Yaping Sun, Lingyun Wang

**Affiliations:** Section of Infectious Diseases, Department of Internal Medicine, Yale University School of Medicine, New Haven, CT 06510, USA; yaping.sun@yale.edu

**Keywords:** HIV, cART, gene editing, CRISPR

## Abstract

The efforts to discover HIV therapeutics have continued since the first human immunodeficiency virus (HIV) infected patient was confirmed in the 1980s. Ten years later, the first HIV drug, zidovudine (AZT), targeting HIV reverse transcriptase, was developed. Meanwhile, scientists were enlightened to discover new drugs that target different HIV genes, like integrase, protease, and host receptors. Combination antiretroviral therapy (cART) is the most feasible medical intervention to suppress the virus in people with HIV (PWH) and control the epidemic. ART treatment has made HIV a chronic infection rather than a fatal disease, but ART does not eliminate latent reservoirs of HIV-1 from the host cells; strict and life-long adherence to ART is required for the therapy to be effective in patients. In this review, we first discussed the scientific history of conventional HIV drug discovery since scientists need to develop more and more drugs to solve drug-resistant issues and release the side effects. Then, we summarized the novel research technologies, like gene editing, applied to HIV treatment and their contributions to eliminating HIV as a complementary therapy.

## 1. Introduction

It has been four decades since the first human immunodeficiency virus (HIV) was identified as the pathogen that caused the disease of acquired immunodeficiency syndrome (AIDS) [1,2]. Since then, scientists have started working on HIV mechanisms and therapeutics. Until now, over thirty HIV drugs have been approved by the Food and Drug Administration (FDA) [3]. Based on their molecular mechanism and targets to each step of the viral life cycle, these drugs were classified into six different groups: (1) coreceptor inhibitors (CRIs) and (2) fusion inhibitors (FIs) targeting viral entry; (3) nucleoside reverse transcriptase inhibitors (NRTIs) and (4) non-nucleoside reverse transcriptase inhibitors (NNRTIs) targeting reverse transcription; (5) integrase strand transfer inhibitors (InSTIs) targeting viral integration; and (6) protease inhibitors (PIs) targeting viral maturation [4,5]. The first generation of drugs is all reverse transcriptase inhibitors, like the first HIV drug, zidovudine (3′-azido-3′-deoxythymidine, or AZT), which was approved by the FDA in 1987 [6,7]. However, AIDS lethality did not decrease until the advent of antiretroviral therapy (ART) in the middle 1990s, after the protease inhibitors were developed.

In general, ART is the combination of three or more drugs designed against at least two steps in the life cycle of the virus. People with HIV (PWH) should be treated with ART as soon as possible. Usually, the initial ART drugs should contain InSTIs, either first-generation InSTIs like raltegravir (RAL) and elvitegravir (EVG) or the second-generation InSTIs like dolutegravir (DLG) and bictegravir (BIC), to specifically block the viral DNA integrated into the host genome [8,9]. The clinical trial of ART showed that HIV viral replication was strongly suppressed, and the plasma viral load of PWH was significantly reduced to an undetectable level [10,11]. There are some clinical studies that demonstrated that no transmissions happened to the partners during condomless intercourse, no matter whether during heterosexual acts or among men who have sex with men (MSM), when the undetectable viral load was maintained for more than six months with ART [12,13,14], which supports the statement that an undetectable viral load equals an untransmittable one (Undetectable = Untransmittable, or U = U) [15,16]. Eventually, ART development has successfully changed AIDS from a fatal disease into a controlled viral infection with restored immune function [17,18] and with the mortality rates of PWH close to general mortality rates [19,20,21]. 

The Department of Health and Human Services Guidelines Panel for the Use of Antiretroviral Agents in Adults and Adolescents with HIV (the Panel) has developed recommendations for the use of statin therapy in people with HIV [22]. Despite the fact that current ART can durably suppress HIV viremia, the viral load will rebound within several weeks of ART interruption, even in patients with very small HIV reservoirs and minimal ongoing viral transcription [23]. The lifelong treatment of ART for PWH has brought new challenges, such as side effects, including weight gain, anxiety, irritability, dizziness, insomnia [24], and drug resistance [25,26]. In addition, ART was unable to eradicate the integrated virus in the host cells [27,28]. The persistence of HIV viral reservoirs in the host cells is always a threat to PWH. Therefore, continuous efforts are demanded to discover novel drugs against different targets with less toxicity and develop alternative therapeutics, such as gene-editing technologies, which may be used to eliminate viral reservoirs in patients and lead to an HIV functional cure. 

In this review, we provide an overview of the HIV-1 life cycle and summarize the most updated anti-HIV drugs that the US Food and Drug Administration (FDA) has approved, according to the viral replication life cycle. In addition, we describe the most advanced gene editing tools based on the clustered regularly interspaced short palindromic repeat (CRISPR)/associated nuclease 9 (Cas9) system, which can be a complementary strategy for HIV cure. Finally, the effective in vivo delivery system for programmable elements will also be briefly discussed.

## 2. HIV Life Cycle and Correlated Target for Drug Discovery

### 2.1. Life Cycle of HIV-1

The HIV-1 life cycle can be viewed as a series of steps processed in order, with some events that may happen simultaneously. The first step is the attachment of viral particles to the host cell membrane by a specific interaction between HIV glycoprotein gp120 and the host cell receptor CD4, which is expressed on the cell surface [29]. Under the observation of variable tropism displayed on HIV-infected CD4^+^ cells, two coreceptors, α-chemokine receptor CXCR4 and β-chemokine receptor CCR5, have been identified as critical factors for T cell (T-tropic) and macrophage (M-tropic) infections, respectively [30,31]. After receptor binding, the membrane was fused under a serial conformational change of glycoprotein gp41. The exposed fusion peptide inserts into the target cell membrane; in the meantime, the N-terminal heptad repeat (NHR) and the C-terminal heptad repeat (CHR) form the coiled-coil six-helix bundle (6HB) bend HIV viral membrane closer to the host cell membrane, which forces the membrane fusion [32]. As a result, viral RNA is released into the host cell cytoplasm.

Once viral RNA is released into the cell cytoplasm, it starts the reverse transcription to generate viral DNA using the viral genome-encoded enzyme reverse transcriptase (RT), also known as RNA-dependent DNA polymerase [33]. The synthesized DNA is associated with several viral and cellular proteins, forming a large nucleoprotein complex called the pre-integration complex (PIC) [34]. The viral DNA, as part of PIC, was transported into the cell nucleus. The viral genome-encoded integrase (IN) catalyzes viral DNA integrated into the host cell genome. The integrated viral DNA, referred to as the “provirus”, is essential for HIV viral replication. It serves as the template for viral RNA synthesis, and the transcribed mRNA is then transported into the cytoplasm and translated to viral proteins. The viral envelop glycoprotein (Env) and the precursor of the Gag protein, Gag-pol polyprotein, are synthesized in the endoplasmic reticulum (ER) and transported to the plasma membrane, where viral particle assembly takes place [35]. During the transportation of viral protein to the membrane, the Gag precursor recruits two copies of single-strand viral RNA, interacts with the Gag-pol precursor, and assembles as viral particles [36]. The assembled Gag protein complex induces membrane budding, and then the viral Env protein is incorporated into the particles [37]. During or immediately after the budding, viral protease (PR) cleaves Gag and the Gag-pol polyprotein precursors to the mature Gag and Pol proteins [38]. The viral particles pinch off from the plasma membrane and are able to initiate a new round of infection. 

### 2.2. Viral Entry Inhibitors 

Blocking HIV infection at the viral entry step is essential to ART. To date, there are three entry inhibitors approved by the FDA, as listed in Table 1 [39]. In 2003, enfuvirtide, also known as T20 or ENF (trade name Fuzeon), was the first HIV entry inhibitor to be approved [40]. Enfuvirtide is a short peptide derived from HIV gp41 amino acids 127–162; it binds to viral native NHR to prevent the formation of a 6HB fusion core and inhibit membrane fusion [41]. However, enfuvirtide is clinically treated at high doses due to its relatively low antiviral activity and short half-life in vivo [42]. Another drawback of enfuvirtide is that it easily induces drug resistance because of viral mutations in NHR [43]. The second entry inhibitor is maraviroc (also known as MVC, trade name Selzentry), which was approved in 2007 as a potent antiviral drug for R5-tropic HIV-1 [44]. Maraviroc is an antagonist of CCR5 and inhibits the binding of chemokine ligands to CCR5 receptors at a low nanomolar range and blocks downstream CCR5-signaling [45]. Maraviroc is a small molecule that can be distributed throughout the body; it can penetrate the blood–brain barrier and can be detected in cerebrospinal fluid (CSF) [46], seminal plasma (SP) [47], and cervicovaginal fluids (CF) [48]. In clinical settings, the resistance of maraviroc appears when HIV switches coreceptor from CCR5 to CXCR4 or to a dual tropism under the pressure of drug treatment [49]. The last entry inhibitor is ibalizumab-uiyk (also known as IBA, trade name Trogarzo), which the FDA approved in 2018 as an HIV treatment drug for adult patients for whom other drugs cannot work [50]. Ibalizumab-uiyk is the only monoclonal antibody (MAb) developed for HIV treatment with intravenous (i.v.) injection over decades. Ibalizumab-uiyk is a recombinant humanized IgG4 MAb, which is derived from a parent mouse MAb. It binds to the interface between extracellular domains 1 and 2 of the human CD4 receptor that induces steric hindrance to prevent HIV gp120 from binding to the CD4 molecule [51]. The advantages of ibalizumab-uiyk for HIV therapy are low toxicity, the ability to restore CD4^+^ T cells, and minimal potential resistance. CCR5 serves as a coreceptor of HIV entry and plays a key role in HIV therapeutics since five out of six HIV patients are cured due to receiving donor hematopoietic cells that present the CCR5 Δ32/Δ32 mutation. The most recent study showed that JAK/STAT impacts CCR5 expression [52]; this may be a new target strategy for developing HIV drugs. 

### 2.3. Reverse Transcriptase Inhibitors

The current FDA-approved anti-HIV drugs for post-entry inhibition target the viral genome-encoded enzymes RT, IN, and PR (Table 1). In fact, more than half of the approved drugs are RT inhibitors. These small molecules can block the viral RNA conversion into DNA, which will be integrated into the host genome later. The first effective anti-HIV drug that the FDA approved in 1987 was an RT inhibitor, zidovudine, also known as azidothymidine (AZT) [7]. Approved RT inhibitors can be grouped into two classes: nucleoside reverse transcriptase inhibitors (NRTIs) and non-nucleoside reverse transcriptase inhibitors (NNRTIs). NRTIs are deoxynucleotide triphosphate analogs but lack the free 3′-hydroxyl group. Once NRTIs are incorporated with the nascent viral DNA, RT enzyme-catalyzed viral DNA synthesis is effectively terminated [51]. For these inhibitors, tenofovir, also known as tenofovir disoproxil fumarate (TDF), is part of the recommended initial ART regimes in clinical settings [53]. These recommended regimes usually have a high rate of viral suppression, minimal toxicity, and a lower risk of drug resistance. NNRTIs are small hydrophobic compounds that bind to an allosteric site located approximately 10 Å away from the RT catalytic active site to inhibit DNA polymerization [54]. Until now, there have been six HIV-1 NNRTIs to be approved by the FDA in the following chronological order: nevirapine (NVP), delavirdine (DLV), efavirenz (EFV), etravirine (ETR), rilpivirine (RPV), and doravirine (DOR). NNRTIs are an essential component in ART and are currently under extensive development. Compared with NRTIs, NNRTIs block viral RNA reverse transcription at the initial steps [55]. However, the emerging drug resistance of NNRTIs makes it fail to beat the increased number of mutated HIV-1 variants. For example, efavirenz, nevirapine, and delavirdine could effectively inhibit wild-type HIV-1 in clinical settings, but they are less effective against RT mutant HIV-1, such as clinical strains containing K103N and Y181C mutations [56]. A recent study by Bertagnolio et al. has shown the rising levels of pretreatment HIV drug resistance among people initiating NNRTIs-based ART [57]. It was confirmed with a similar study from Hauser et al., which reported drug resistance mutations among adults under NNRTI-based ART in Southern Africa [58]. By using a structure-based design for new compounds, Kang et al. synthesized two piperidine-substituted thiophene; both compounds exhibited increased anti-HIV potency against NNRTI-resistant strains [59]. It is encouraging that structure modification on the currently available compounds could achieve improved anti-HIV potency.

### 2.4. Protease Inhibitors

It has been studied that the proteolytic cleavage of Gag and the Gag-pol polyprotein precursors by HIV PR is required for viral infectivity. The budded uncleaved virions lose the infection ability [60]. Consequently, the critical role of the PR enzyme in the process of virus maturation makes it an attractive target for anti-HIV drugs. So far, there have been ten FDA-approved protease inhibitors (PIs), including saquinavir, indinavir, ritonavir, nelfinavir, amprenavir, lopinavir, fosamprenavir, atazanavir, tipranavir, and darunavir (Table 1). The first one, saquinavir (SQV), was approved in 1995, which marked the beginning of combination antiretroviral therapy in HIV patients. Clinical data has shown that ART with saquinavir and RT inhibitor zalcitabine significantly extended patient lifespan compared with zalcitabine alone [61]. Protease inhibitors have thus become one of the most important regimens in combination therapy. Unfortunately, most of the PIs are associated with side effects in long-term treatment, such as the increased risk of cardiovascular and cerebrovascular diseases, as well as dyslipidemia and diabetes [62,63]. The adverse effects were not significant when comparing PI as a monotherapy compound to the ART combination of PIs and RT inhibitors, suggesting that PIs were major responsibility for the side effects. 

It is possible to optimize the chemical structure of PI molecules to improve the clinical benefits. The successful attempts are darunavir and lopinavir, which were modified from amprenavir and ritonavir, respectively. However, further modification of lopinavir failed, and none of the analog compounds showed better results than the original lopinavir [64]. Other attempts have been made to optimize PIs to avoid side effects. For example, GS-8374, modified from a scaffold of TMC-126 (darunavir analog), has a favorable resistance profile against a spectrum of patient-derived HIV-1 variants and is highly resistant to multiple PIs [65]. In addition, this new inhibitor GS-8374 neither affects insulin-stimulated glucose uptake in adipocytes in culture nor acutely alters peripheral glucose disposal in a rodent model system as preclinical evaluation, which is similar to atazanavir but unlike ritonavir and lopinavir [66]. 

### 2.5. Integrase Inhibitors

Like all retroviruses, HIV-1 integrates its RNA reverse-transcribed DNA into the host cell chromosome. Integration provides a favorable environment for long-term virus persistence. DNA integration is mediated by the viral genome-encoded enzyme IN, which is a specialized DNA recombinase. IN is an attractive drug target because it is essential for infective virion production, and there is no mammalian homolog of IN [67]. However, the discovery of anti-IN drugs was delayed for over 10 years compared with drugs targeting RT and PR. This is mainly because of a lack of good lead compounds obtained from in vitro screening and the fact that there were no reliable assays to evaluate integrase inhibition. Until now, there have been five anti-IT compounds that were approved by the FDA (Table 1), including the first IN inhibitor, raltegravir, which was approved in 2007, and elvitegravir, dolutegravir, bictegravir, and cabotegravir. 

After HIV RNA is reverse-transcribed to viral DNA, IN assembles at its ends by binding to the HIV-1 LTR region, which is a consequence of the DNA–IN binding complex formation. Then, IN catalyzes to remove two terminal nucleotides at each end of LTRs to produce a new 3′ hydroxyl ends, a process referred to as 3′ processing [68]. At the second catalytic step, IN is responsible for transferring viral DNA to the human chromosomal DNA, namely strand transfer or transesterification. IN binds to the host chromosomal DNA and mediates a concerted nucleophilic attack by the 3′ hydroxyl residues of the viral DNA on phosphodiester bridges located in the target DNA. Then, the processed 3′ hydroxyl ends of viral DNA are ligated to the 5′-O-phosphate ends of the host DNA, irreversibly binding the viral DNA to the target DNA [69]. The last step of integration is filling the gap between virus and host DNA by host repair machinery. The current approved InSTIs were initially developed to target the strand transfer step of HIV-1 integration, and it is the only category that interacts with two essential elements of the virus in the DNA–IN complex [70]. InSTIs were well tolerated by PWH, but emerging data suggest that some InSTIs contribute to weight gain. Because of its high efficacy and fewer side effects, InSTIs are recommended as an initiating ART component [71].

## 3. CRISPR/Cas9-Based Gene Editing in HIV Treatment

### 3.1. Gene Editing Targets Host Cell

It has been reported that the 32-base-pair deletion of coreceptor CCR5 renders host cells resistant to R5-tropic HIV-1 variants infection [72]. In 2008, at the CROI (Conference on Retroviruses and Opportunistic Infections) conference, the Berlin patient was declared the first person “cured of HIV”, who remained free of HIV for more than 13 years without ART after stem cell transplantation from a healthy donor with homozygous CCR5Δ32 mutation. After the Berlin patient, there have been the London patient, the New York patient, the City of Hope patient, and the Düsseldorf patient, who have been cured of HIV from hematopoietic stem cell (HSC) transplantation [73]. Therefore, intensive work on the disruption of CCR5 expression with gene-editing tools is currently being conducted. The rapidly updated CRISPR/Cas9 technology has boosted the research for curative HIV therapy, such as base editing. Base editing uses programmable DNA-binding proteins to directly convert C⋅G to T⋅A base pairs (cytosine base editing, CBE) [74] or convert C⋅G to T⋅A base pairs (adenine base editing, ABE) [75]. Base editors modify targeted nucleotides with specific base changes in the genome (Figure 1), while CRISPR/Cas9 breaks double-strand and randomly modifies the genomic DNA. 

Xu et al. established an HSC transplantation model with CRISPR/Cas9-conferred CCR5-ablated human CD34^+^ hematopoietic stem/progenitor cells (HSPCs) [76]. HSPCs were reconstituted in mice for over 1 year and achieved robust CCR5 disruption, which mediated an HIV-1 resistance effect in vivo. Liu et al. designed two different guide RNA (gRNA) combinations targeting both CXCR4 and CCR5 in a single vector [77]. The simultaneous genome editing of HIV-1 coreceptors CXCR4 and CCR5 in primary CD4^+^ T cells with the CRISPR/Cas9 system protects modified cells from X4-tropic or R5-tropic HIV-1 viral infection. Knipping et al. applied base editors to simultaneously disrupt both coreceptors in primary human CD4^+^ T cells; it prevents transduction with R5-tropic and X4-tropic viral vectors [78]. Except for targeting coreceptors, Chinnapaiyan et al. used CRISPR/Cas9 to knock down the cellular co-factor cyclin T1, which is crucial for HIV transcription and demonstrated cyclin T1 inhibition-mediated HIV silencing [79]. Overall, the research on CCR5 gene modification is dominant over other host cellular targets. 

### 3.2. Gene Editing Targets HIV Genome

Although the current HIV-1 treatment of ART and broadly neutralizing antibodies could suppress plasma viral load below the detectable level, ART cannot eliminate the integrated provirus from host cells. Viral load will rebound within a few weeks after ART withdrawal. This challenge can be overcome with gene editing on proviral DNA in CD4^+^ T cells. Ebina et al. designed gRNA to target HIV-1 long terminal repeat (LTR) in specific locations of the TAR sequence of the R region and NF-κB binding sequence in the U3 region, respectively, and this LTR-targeted CRISPR/Cas9 system can disrupt HIV-1 provirus and active the provirus from the cellular genome [80]. Kaminski et al. employed a CRISPR/Cas9 editing system to precisely remove the integrated copies of the proviral DNA fragment from latently infected human CD4^+^ T cells by targeting the highly conserved sequence of U3 LTR region of all isolated HIV-1 strains [81]. Liao et al. adapted the CRISPR/Cas9 system to disrupt the latently integrated viral genome and provide long-term defense against new viral infections. They screened several potential gRNA targeting sites in the HIV-1 genome, including the structural (gag and env), enzymatic (pol) and accessory genes (vif and rev), as well as LTRs, and they increased the frequency of disruption and activation of the pre-integrated proviral genome by using a multiplexed CRISPR/Cas9 system along with the gRNAs targeting LTR sequences (especially the R region) [82]. Zhu et al. tested 10 sites in HIV-1 DNA using CRISPR/Cas9 and found that a highly efficient target site is located in the second exon of rev, and it could inactivate provirus efficiently by significantly reducing HIV-1 gene expression and virus production [83]. In general, among different sites targeted on the viral genome, targeting HIV-1 LTR has a strong impact on viral disruption because LTR serves as a critical element of the viral transcription, especially targeting the LTR-R region, which contains the TAR sequence, which is highly conserved of all HIV-1 subtypes. 

In addition, there are combinational targets on both host cells and the viral genome. To improve the efficiency of viral elimination, Dash et al. developed dual CRISPR, which targets both proviral DNA and CCR5 [84]. There are two CRISPR reagents, including one set designed to target LTR and Gag to activate the HIV-1 LTR-Gag region from latent proviral DNA integrated cells and the other set designed to target host coreceptor CCR5. The viral outgrowth assay (VOA) demonstrated that no progeny virus was recovered in plasma and tissues from CRISPR-treated virus-free mice. 

### 3.3. Clinical Trials of Gene Editing Applied in HIV Treatment 

On 8 December 2023, the U.S. FDA approved two milestone treatments, Casgevy and Lyfgenia, for sickle cell disease (SCD) in patients 12 years and older [85]. Casgevy is the first FDA-approved therapy utilizing CRISPR/Cas9, which greatly inspired the development of HIV treatment in the field of gene therapy. To date, EBT-101 is the only CRISPR/Cas9-based gene therapy for HIV treatment in clinical trials and is now in the status of recruiting (NCT05144386). EBT-101 is comprised of an all-in-one CRISPR/Cas9 system that expresses dual gRNAs targeting viral LTRs and the Gag gene, thereby generating three possible deletions: 5′LTR to Gag, Gag to 3′LTR, and 5′LTR to 3′LTR [86]. The dual gRNAs excise large sections of proviral DNA (Figure 2), eliminating viral escape and reproduction.

Furthermore, there are zinc finger nucleases (ZFNs) that enable CCR5 gene editing for HIV treatment in clinical trials. A phase I study of autologous T cells genetically modified to target CCR5 using ZFNs was completed (NCT00842634). It demonstrated that the loss of CCR5 protein on the T cells is permanent in the follow-up study up to 6 years. Meanwhile, other phase I (NCT01044654) and phase 1/2 (NCT01252641) clinical trials regarding the dose escalation and single-dose infusion of autologous T cells genetically modified to target CCR5 using ZFNs in HIV-infected patients were completed at almost the same time. Both of the trials proved the safety and feasibility of ZFN-CCR5-modified CD4^+^ T cells re-infused back to humans as HIV therapy. Further, the phase II clinical trial that is randomly comparing the effect of infusing expanded autologous CD4^+^ T cells with or without CCR5 modification ex vivo using ZFNs among HIV-infected patients is ongoing (NCT03666871). In addition, there is a phase I pilot study of stem cell gene modification, which is evaluating the feasibility, safety, and engraftment of ZFN-CCR5-modified CD34^+^ HSPCs in R5 tropic HIV-1 infected patients (NCT02500849). 

Although gene editing is a robust tool for HIV treatment, balancing the potential benefits and ethical considerations is important. DNA modification on reproductive embryos is banned in many countries; it is urgently needed to create ethical guidelines and make them transparent. Public engagement is crucial, making the patient group aware of gene editing technology and its pros and cons by involving communication between scientists and patients [87]. Additionally, the long-term effects and potential unintended edits on the human genome still require extensive study to address the issue. It is essential to avoid off-target effects to ensure maximum benefits and minimum risks in HIV gene therapy [88]. 

## 4. Conclusions and Future Perspectives

The discovery of anti-HIV drugs is arguably among the most successful achievements for any human disease, considering the number of available anti-HIV agents that have been developed for four decades since the first HIV-1 viral infection case was confirmed in 1981. More than 30 antiretroviral drugs have been approved, and combination therapy has been demonstrated with high efficiency and controllable toxicity for PWH. However, the lifelong treatment of ART and acquired drug resistance are still the key issues in HIV cure. Continuous efforts are demanded to develop new compounds and new drug combinations to achieve therapy success. Although these ART drugs highly suppress the viremia, they are unable to eradicate the integrated viral DNA. With the development of gene-editing tools, such as ZFNs, CRISPR/Cas9, and transcription activator-like effectors (TALENS) etc., more and more research has been conducted on provirus elimination using these new technologies. Especially in recent years, base editing and prime editing, which are derived from the CRISPR/Cas9 system, are much safer compared with classical CRISPR/Cas9 since these two editing methods do not break double-strand targeted DNA. With low off-target effects, these gene modification tools can be applied in various research fields, including anti-HIV therapy. It is convenient and only needs to design different gRNA for different targets. Moreover, it is flexible, and different gRNAs can be combined in one vector for better knock-out efficiency. For example, the current ongoing clinical trial of EBT-101 is a combination of gRNAs targeting HIV-1 LTR and Gag genes. 

The CRISPR/Cas9 system is indeed a promising tool applied for HIV treatment, but an obstacle is its low delivery efficiency in vivo. The delivery components can be plasmid, ribonucleoprotein (RNP), and mRNA. The delivery vehicle can be viral vector-based delivery, lipid nanoparticle, and microinjection. Using nanoparticles to deliver ART drugs could maintain effective drug concentrations in targeted tissues for HIV treatment [89]. Adeno-associated virus (AAV) vector is an efficient and widely used delivery agent, and it is the only gene therapy vector that the FDA has approved for human diseases. Viral vector delivery for gene-editing elements also made remarkable progress in clinical HIV treatment. Specifically, EBT-101 adapts AAV9 for CRISPR/Cas9 delivery. Hamann et al. modified the AAV2 capsid protein with a set of novel nanobodies with high affinity for the human CD4 receptor. This CD4 antibody-modified capsid was demonstrated to improve the targeting efficiency of human primary CD4^+^ T cells in vitro [90]. It provides a promising strategy for changing AAV tropism to particularly targeting specific cells. The limitation of the AAV vector is its packaging capacity, which is less than 5 kb. The large size of SpCas9 (~4.1 kb) decreases the efficiency of delivery, so it seems impossible to package the base editor and prime editor in AAV. Therefore, Levy et al. came up with a dual AAV system for the delivery of split-base editors [91]. Another way to solve the issue is to optimize the SpCas9 protein into a truncated variant or use a smaller SaCas9. While there is a long way to go, the combination of antiretroviral drugs and gene-editing technology will lead to promising progress in HIV-1 therapeutics. 

## Figures and Tables

**Figure 1 pharmaceuticals-17-00887-f001:**
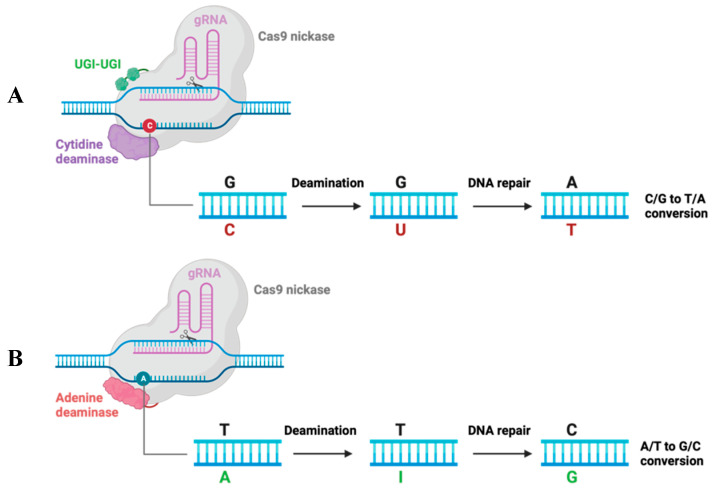
Mechanism schematic of the cytidine base editor (CBE) and adenine base editor (ABE) systems. (**A**) CBE contains a Cas9 nickase fused to a deaminase and two UGIs (uracil glycosylase inhibitors). C changed to U under deamination, then changed to T with host DNA repair. As a result, CBE converts the C/G base pair into the T/A base pair. (**B**) ABE is composed of a Cas9 nickase fused to deaminase that changes A to I, then I to G. As a result, ABE converts the A/T base pair into a G/C base pair.

**Figure 2 pharmaceuticals-17-00887-f002:**
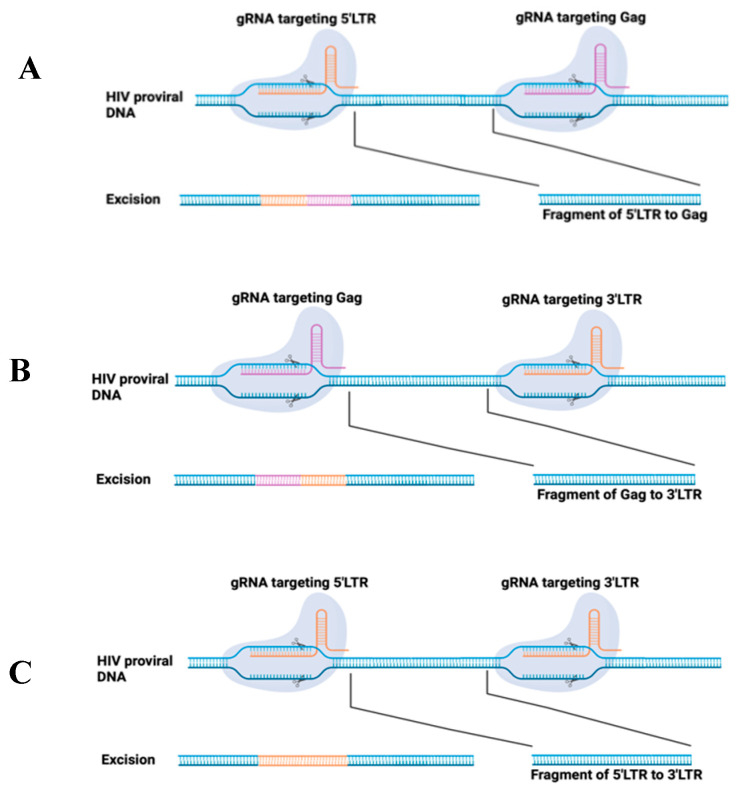
Schematic of CRISPR/Cas9 system applied for HIV provirus excision. Two gRNAs were designed to target HIV LTR (5′ and 3′) and Gag at specific sites. The induced double-strand breaks are repaired using non-homologous end joining, which will induce the deletion of a fragment from 5′LTR to Gag (**A**), or Gag to 3′LTR (**B**), or 5′LTR to 3′LTR (**C**).

**Table 1 pharmaceuticals-17-00887-t001:** List of FDA-approved HIV drugs by targets.

Year	Trade Name	Generic Name	Molecule Type	Target	Manufacturer	Class
2003	Fuzeon	Enfuvirtide (T20)	Peptide	GP41	Trimeris/Roche	FI
2007	Selzentry	Maraviroc (MVC)	Small molecule	CCR5	Pfizer	CRI
2018	Trogarzo	Ibalizumab-uiyk (IBA)	monoclonal antibody	CD4	TaiMed Biologics	RI
1987	Retrovir	Zidovudine (AZT)	Small molecule	RT	GlaxoSmithKline	NRTI
1991	Videx	Didanosine (ddI)	Small molecule	RT	Bristol-Myers Squibb	NRTI
1992	Hivid	Zalcitabine (ddC)	Small molecule	RT	Roche	NRTI
1994	Zerit	Stavudine (d4T)	Small molecule	RT	Bristol-Myers Squibb	NRTI
1995	Epivir	Lamivudine (3TC)	Small molecule	RT	GlaxoSmithKline	NRTI
1996	Viramune	Nevirapine (NVP)	Small molecule	RT	Boehringer Ingelheim	NNRTI
1997	Rescriptor	Delavirdine (DLV)	Small molecule	RT	ViiV Healthcare	NNRTI
1998	Sustiva	Efavirenz (EFV)	Small molecule	RT	DuPont Pharmaceuticals	NNRTI
1998	Ziagen	Abacavir (ABC)	Small molecule	RT	ViiV Healthcare	NRTI
2001	Viread	Tenofovir disoproxil (TDF)	Small molecule	RT	Gilead	NRTI
2003	Emtriva	Emtricitabine (FTC)	Small molecule	RT	Gilead	NRTI
2008	Intelence	Etravirine (ETR)	Small molecule	RT	Johnson & Johnson	NNRTI
2011	Edurant	Rilpivirine (RPV)	Small molecule	RT	Tibotec	NNRTI
2018	Pifeltro	Doravirine (DOR)	Small molecule	RT	Merck	NNRTI
1995	Invirase	Saquinavir (SQV)	Small molecule	PR	Roche	PI
1996	Crixivan	Indinavir sulfate (IDV)	Small molecule	PR	Merck	PI
1996	Norvir	Ritonavir (RTV)	Small molecule	PR	Abbott	PI
1997	Viracept	Nelfinavir (NFV)	Small molecule	PR	Agouron Pharmaceuticals	PI
1999	Agenerase	Amprenavir (APV)	Small molecule	PR	GlaxoSmithKline	PI
2000	Kaletra	Lopinavir (LPV)	Small molecule	PR	Abbott	PI
2003	Lexiva	Fosamprenavir (FPV)	Small molecule	PR	GlaxoSmithKline	PI
2003	Reyataz	Atazanavir (ATV)	Small molecule	PR	Bristol-Myers Squibb	PI
2005	Aptivus	Tipranavir (TPV)	Small molecule	PR	Boehringer Ingelheim	PI
2006	Prezista	Darunavir (DRV)	Small molecule	PR	Tibotec	PI
2007	Isentress	Raltegravir (RAL)	Small molecule	IN	Merck	InSTI
2013	Tivicay	Dolutegravir (DTG)	Small molecule	IN	ViiV Healthcare	InSTI
2014	Vitekta	Elvitegravir (EVG)	Small molecule	IN	Gilead	InSTI
2018	Biktarvy	Bictegravir (BIC)	Small molecule	IN	Gilead	InSTI
2021	Vocabria	Cabotegravir (CAB)	Small molecule	IN	ViiV Healthcare	InSTI

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
