# Peer review of "Development of Anti-HIV Therapeutics: From Conventional Drug Discovery to Cutting-Edge Technology"

_pharmaceuticals, 2024, doi:10.3390/ph17070887_

Round 1

Reviewer 1 Report

Comments and Suggestions for Authors

In the manuscript, the drugs used in Anti-HIV Therapeutics are mentioned. All drug groups, from traditional to the latest technology, are explained in detail. The review is quite well designed.

line 88, "tranported" please check spelling.

General side effects of drug groups can be added.

Author Response

Dear reviewer,

This is point-by-point response to Comments and Suggestions:

In the manuscript, the drugs used in Anti-HIV Therapeutics are mentioned. All drug groups, from traditional to the latest technology, are explained in detail. The review is quite well designed. (Thanks for your comment!)

line 88, "tranported" please check spelling. (Thanks! I have corrected it.)

General side effects of drug groups can be added. (Thanks! I have added the general side effects reported in the literature as described in line 59.)

Reviewer 2 Report

Comments and Suggestions for Authors

1.      Discussing standard two-drug, three-drug, and four-drug regimens and fixed-dose combinations is missing,

2.      Moreover, the authors fail to discuss lenacapavir (CA inhibitor), islatravir (NRTTI) and anti-gag compounds.

3.      In addition, please make new paragraph about the strategies for eradication of HIV-1.

4.      Guidelines for the use of antiretroviral agents in adults and adolescents with HIV should be included.

5.      Opportunities and challenges in new HIV therapeutic discovery should be discussed.

6.      The authors omitted any comments about the toxic effect of Anti-HIV drug and their management strategies.

7.      Please provide some details about the targeted drug delivery of anti-HIV drugs.

8.      Use of nanotherapeutics as anti-HIV drug should be mentioned.

9.      What is the novelty of your work compared to the others.

10.  How is the etiology of HIV characterized across different populations?

11.  What detailed methodology is used in the original research sources to evaluate efficacy, onset of action and safety of anti-HIV over time.

Comments on the Quality of English Language

Syntax and grammar should be double checked.

Author Response

Dear reviewer,

This is point-by-point response to Comments and Suggestions:

1.      Discussing standard two-drug, three-drug, and four-drug regimens and fixed-dose combinations is missing, (Thank you for the comment! The related content is written in the second paragraph of Introduction from line 38 to 43. For the doses, it is too specific with the drugs, so we have not discussed about it in the manuscript due to the word limitation of this review.)

2.      Moreover, the authors fail to discuss lenacapavir (CA inhibitor), islatravir (NRTTI) and anti-gag compounds. (Thank you for the comment! We discussed some of the compounds as representative but not all of them due to the word limitation of this review.)

3.      In addition, please make new paragraph about the strategies for eradication of HIV-1. (Thank you for the comment! The related content is written in the section of “3.2. Gene Editing Targets HIV Genome” from line 279 to 282 and in the section of “3.3. Clinical Trials of Gene Editing Applied in HIV Treatment” from line 309 to 313.)

4.      Guidelines for the use of antiretroviral agents in adults and adolescents with HIV should be included.(Thank you for the comment! We have added it in the Introduction at line 53-55.)

5.      Opportunities and challenges in new HIV therapeutic discovery should be discussed. (Thank you for the comment! We have added the content about opportunities of new HIV drug development at the end of section “2.3. Reverse Transcriptase Inhibitors”, we have added the content about challenge of gene therapy at the end of section 3.)

6.      The authors omitted any comments about the toxic effect of Anti-HIV drug and their management strategies. (Thank you for the comment! We have added the side effects of ART drugs in the Introduction part at line 59.)

7.      Please provide some details about the targeted drug delivery of anti-HIV drugs. (Thank you for the comment! We have mentioned it in section 4. Conclusions and Future Perspectives at line 370 to 373.)

8.      Use of nanotherapeutics as anti-HIV drug should be mentioned. (Thank you for the comment! We have added it in section 4. Conclusions and Future Perspectives at line 365 to 366.)

9.      What is the novelty of your work compared to the others. (Thank you for the comment! We think we have systematically summarized all the currently available ART drugs, most importantly we have discussed about the conventional CRISPR/Cas9 that has entered clinical trial, which could potentially remove latent HIV DNA from host cells. We have discussed the most recent base editor applied for HIV gene therapy.)

10.  How is the etiology of HIV characterized across different populations? (Thank you for the question! HIV genome sequence diversity not only happened across different population but also at different stage of infection. HIV infection is characterized by rapid and error-prone viral replication resulting in genetically diverse virus populations. Diversity accumulated in recently infected individuals at rates 30-fold higher than in patients with chronic infection. Several parameters potentially influencing HIV-1 transmission, eg, concomitant sexually transmitted infections (STIs) and mode of transmission. The related literatures are listed here: PMID: 23678164; PMID: 21998286)

11.  What detailed methodology is used in the original research sources to evaluate efficacy, onset of action and safety of anti-HIV over time. (Thank you for the question! To evaluate efficacy and safety of ART drug in clinic, viral suppression and CD4 outcomes were frequently reported at multiple timepoints and were analyzed separately for each of the three timepoints of interest: 24 weeks, 48 weeks, and 96 weeks. For viral suppression, the principal analysis used various thresholds, with preference for less than 50 copies per mL. See https://clinicaltrials.gov/study/NCT05911360 and PMID: 27658869.)

Comments on the Quality of English Language

Syntax and grammar should be double checked. (Thanks for your suggestions! We have checked the manuscript thoroughly.)

Reviewer 3 Report

Comments and Suggestions for Authors

The manuscript provides a comprehensive review of the development of anti-HIV therapeutics, tracing the evolution from traditional drug discovery methods to modern gene editing technologies. The paper is well-structured, covering key aspects of HIV therapeutic development in a logical order. However, there are several areas where improvements can be made, particularly concerning grammar and typographical errors, the organization of Table 1, and the depth of discussion on certain topics.

But some problems need to be addressed before publication.

The document contains minor grammatical errors and typographical mistakes.

Table 1 is confusing because some names appear more than once.

Reverse Transcriptase Inhibitors

The section contains minor typographical errors. For instance, the word "blockingk" should be corrected to "blocking". Additionally, the discussion of drug resistance in NNRTIs could be expanded. Correct the typographical error by changing "blockingk" to "blocking". Expand the discussion on drug resistance in NNRTIs to include more recent findings and potential solutions. 

Gene Editing Technologies

The section could be expanded to discuss ethical considerations and the long-term implications of gene editing in HIV treatment. Expand this section to include a discussion on ethical considerations and the long-term implications of gene editing in HIV treatment. 

Comments on the Quality of English Language

The document contains minor grammatical errors and typographical mistakes.

Author Response

Dear reviewer,

This is point-by-point response to Comments and Suggestions:

The manuscript provides a comprehensive review of the development of anti-HIV therapeutics, tracing the evolution from traditional drug discovery methods to modern gene editing technologies. The paper is well-structured, covering key aspects of HIV therapeutic development in a logical order. However, there are several areas where improvements can be made, particularly concerning grammar and typographical errors, the organization of Table 1, and the depth of discussion on certain topics. (Thanks for your comment!)

But some problems need to be addressed before publication.

The document contains minor grammatical errors and typographical mistakes. (Thanks! We have double-checked throughout the manuscript and correct it as much as we can.)

Table 1 is confusing because some names appear more than once. (Thanks! We have corrected the Table 1, please see the updated table in the manuscript.)

Reverse Transcriptase Inhibitors

The section contains minor typographical errors. For instance, the word "blockingk" should be corrected to "blocking". Additionally, the discussion of drug resistance in NNRTIs could be expanded. Correct the typographical error by changing "blockingk" to "blocking". Expand the discussion on drug resistance in NNRTIs to include more recent findings and potential solutions. (Thanks for your comment! We have changed the typographical error for "blocking"; The discussion of drug resistance in NNRTIs is added and included more recent findings and potential solutions as you suggested. Please see the updated information in the manuscript line 172-179.)

Gene Editing Technologies

The section could be expanded to discuss ethical considerations and the long-term implications of gene editing in HIV treatment. Expand this section to include a discussion on ethical considerations and the long-term implications of gene editing in HIV treatment. 

(Thanks for your comment! We have added the information as you suggested at the end of this section.)

Comments on the Quality of English Language

The document contains minor grammatical errors and typographical mistakes. (Thanks for your comment! We have corrected them throughout the manuscript.)

Round 2

Reviewer 3 Report

Comments and Suggestions for Authors

the authors have addressed all my concerns.